# Dyslexia Prediction from Natural Reading of Danish Texts

**Marina Björnsdóttir**
IT University of Copenhagen
University of Copenhagen
marina.bjorns@gmail.com

**Nora Hollenstein**
University of Copenhagen
University of Zurich
nora.hollenstein@hum.ku.dk

**Maria Barrett**
IT University of Copenhagen
mbarrett@itu.dk

## Abstract

Dyslexia screening in adults is an open challenge since difficulties may not align with standardised tests designed for children. We collect eye-tracking data from natural reading of Danish texts from readers with dyslexia while closely following the experimental design of a corpus of readers without dyslexia. To our knowledge, this is the first attempt to classify dyslexia from eye movements during reading in Danish. We experiment with various machine-learning methods, and our best model yields a 0.85 macro F1 score.

## 1 Introduction

Dyslexia is a learning disorder of neurological origin that reportedly affects about 10-20% of the world population (Rello and Ballesteros, 2015; Kaisar, 2020). It involves difficulties with reading, spelling, and decoding words, and is not related to intelligence (Perera et al., 2018; Rauschenberger et al., 2017). Detecting dyslexia as early as possible is vital, as the disorder can lead to many negative consequences that can be mitigated with proper assistance. These include low self-esteem and high rates of depression and anxiety (Perera et al., 2018; Schulte-Körne, 2010). There are qualitative studies suggesting that living with an undiagnosed learning disorder leads to frustrations (Kong, 2012), feelings of being misunderstood (Denhart, 2008), and of failure, (Tanner, 2009). Being diagnosed with a learning disorder as an adult has been reported to lead to a sense of relief (Arceneaux, 2006), validation (Denhart, 2008; Kelm, 2016) and liberation (Tanner, 2009; Kong, 2012). Dyslexia can be difficult to diagnose due to its indications and impairments occurring in varying degrees (Eckert, 2004), and is therefore often recognised as a *hidden disability* (Rello

and Ballesteros, 2015). Popular methods of detecting dyslexia usually include standardised lexical assessment tests that involve behavioural aspects, such as reading and spelling tasks (Perera et al., 2018). Singleton et al. (2009) explain that computerised screening methods have been well-established for children in the UK, but developing such tests for adult readers with dyslexia is exceptionally challenging as adults with dyslexia may not show obvious literacy difficulties that align with what standardised tests distinguish as dyslexic tendencies. For one thing, dyslexia is experienced differently from person to person. Still, also, most adults with dyslexia have developed strategies that help them disguise weaknesses and may thus remain unnoticed and result in false-negative tests (Singleton et al., 2009).

Less frequently used methods are eye tracking during reading or neuroimaging techniques such as (functional) magnetic resonance imaging, electroencephalogram, brain positron emission tomography, and magnetoencephalography methods (Kaisar, 2020; Perera et al., 2018). These models are yet under experimental development and are currently not used for screening dyslexia (Perera et al., 2018). A small body of studies investigates dyslexia detection using eye tracking with the help of machine-learning techniques outlined in §2.4. Compared to neuroimaging techniques, eye tracking is more affordable and faster to record and its link to online text processing is well established (Rayner, 1998). Using eye-tracking records for dyslexia detection does not necessarily require readers to respond or perform a test but merely objectively observes the reader during natural reading (Benfatto et al., 2016). Although eye-tracking experiments are often limited to a relatively small number of participants compared to computerized tools, the method typically produces many data points from each participant.

The purpose of the current paper is twofold:

1) We provide a dataset from participants with dyslexia reading Danish natural texts. This dataset uses the same experimental design as the CopCo corpus by Hollenstein et al. (2022), which allows us to compare the eye movement patterns from readers with dyslexia to those without from CopCo. 2) We train the first machine learning (ML) classifiers for dyslexia prediction from eye movements in Danish. The data is available as raw gaze recording, fixation-level information, and word-level eye tracking features.[1] The code for all our experiments is also available online.[2]

## 2 Related Work

### 2.1 Dyslexia Screening in Denmark

In 2015, The Ministry of Children and Education in Denmark launched a national electronic dyslexia test, Ordblindetesten 'the Dyslexia Test'. The test is a screening method for children, youths, and adults speculated to have dyslexia. It is accessible through educational institutions and is performed under the observation of a supervisor (Centre for Reading Research et al., 2020). It consists of three multiple-choice subtests, performed electronically, that focus on phonological decoding abilities. The result is 'not dyslexic,' 'uncertain phonological decoding,' or 'dyslexic.' The official instruction strictly denies the uncertain group to be dyslexic[3] and therefore not entitled to dyslexia support. But they may benefit from other support and are subject to further assessment, e.g., text comprehension, reading speed, spelling, and vocabulary tests appropriate for the examinee's age and educational requirements (Centre for Reading Research et al., 2020). To this end, Helleruptesten "The Hellerup Test" is used by educational institutions for adults.[4]

### 2.2 Danish as a Target Language

Similar studies on dyslexia detection with ML classification include experiments with Chinese (Haller et al., 2022), Swedish (Benfatto et al., 2016), Spanish (Rello and Ballesteros, 2015), Greek (Asvestopoulou et al., 2019), Arabic (Al-Edaily et al., 2013) and Finnish (Raatikainen et al.,

2021) as their target languages. However, the diagnostic characteristics of dyslexia may differ depending on the transparency of the language. In early research, De Luca et al. (1999) reported that the regular spelling-sound correspondences in languages of transparent orthographies, e.g., German and Italian, dim phonological deficits. Phonological deficits of individuals with dyslexia are clearer in languages with irregular, non-transparent orthographies (Smyrnakis et al., 2017).

Danish is a language with a highly non-transparent orthography. It has been shown that overall adult reading comprehension skills are poorer in Danish than in other Nordic languages (Juul and Sigurdsson, 2005). The lack of spelling-sound correspondence in Danish indicates that the Danish language holds excellent value for investigating dyslexia detection based on two main reasons: Firstly, the combination of the non-transparent orthography of the Danish language and eye movement patterns could potentially reveal more apparent indications of dyslexia through the selected features that have proven to be relevant for dyslexia detection in other languages, which can be favourable in further research on, e.g., the development of assistive tools and technologies. Enabling a direct comparison between eye-tracking data from adults with dyslexia and adults without dyslexia with Danish as the target language will provide beneficial insights into reading dyslexic patterns, which can be favourable in further research, e.g., the development of assistive tools and technologies. Secondly, the fact that reading comprehension skills are proven to be poorer in Danish than in other Nordic languages highlights the necessity of proper assistance and recognition for individuals with dyslexia in Denmark.

### 2.3 Dyslexia and Eye Movements

Tracking eye movements during natural reading reveals information on fixations (relatively still gaze on a single location) and saccades (rapid movements between fixations). Studies (Rayner, 1998; Henderson, 2013) have substantiated that information on eye movements during reading contains characterizations of visual and cognitive processes that directly impact eye movements. These are also strongly related to identifying information about, e.g., attention during reading, which is highly correlated with saccades (Rayner, 1998).

---

[1] https://osf.io/ud8s5/
[2] https://github.com/norahollenstein/copco-processing
[3] https://www.spsu.dk/for-stoettegivere/elever-og-studerende-med-usikker-fonologisk-kodning
[4] from Vestegnen VUC, an educational institution that provides education for students with dyslexia

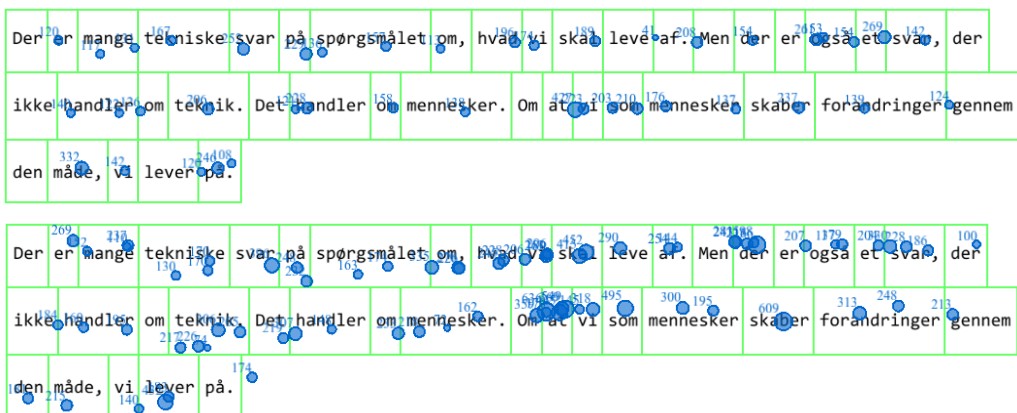

Figure 1: Fixations recorded from a reader without dyslexia (above) and a reader with dyslexia (below) when reading the same sentence. Numbers indicate duration in ms.

As Henderson (2013) phrases it, "eye movements serve as a window into the operation of the attentional system."

Previous studies have repeatedly shown that readers with dyslexia show more fixations and regressions, longer fixation durations, and shorter and more numerous saccades than readers without dyslexia (Pirozzolo and Rayner, 1979; Rayner, 1986; Biscaldi et al., 1998). This was already discovered by Rubino and Minden (1973) and later work discussed whether this was the cause or effect of dyslexia with evidence on both sides, e.g., Pirozzolo and Rayner (1979); Pavlidis (1981); Eden et al. (1994); Biscaldi et al. (1998). Most recent studies acknowledge that the movements reflect a dyslexic reader's difficulties with processing language. (Fischer and Weber, 1990; Hyönä and Olson, 1995; Henderson, 2013; Rello and Ballesteros, 2015; Benfatto et al., 2016; Raatikainen et al., 2021), and Rayner (1998) who echo an earlier study (Rayner, 1986) state that eye movements are not the cause of slow reading but rather reflect the more time-consuming cognitive processes. These insights from psycholinguistics motivate the feature selection for this work.

## 2.4 ML-based Dyslexia Detection from Gaze

Recent evidence shows that ML-based methods can be used for dyslexia detection in children, e.g., Christoforou et al. (2021); Nerušil et al. (2021). This section is, however, limited to ML-based methods for dyslexia detection in adults. Prior studies that facilitate the investigation of dyslexia detection with the help of machine learning classification on eye-tracking data have concluded that support vector machines (SVM's) is of great advantage (Rello and Ballesteros, 2015; Benfatto et al., 2016; Prabha and Bhargavi, 2020; Asvestopoulou et al., 2019; Raatikainen et al., 2021). Rello and Ballesteros (2015) used an SVM for dyslexia detection based on eye-tracking recordings from readers with and without dyslexia, which resulted in an accuracy of 80.18%. Benfatto et al. (2016); Prabha and Bhargavi (2020) achieved accuracy scores of 95.6% and 95% respectively on the same dataset using SVM variations.

With Greek as their target language, Smyrnakis et al. (2017) propose a method with two parameters for dyslexia detection: word-specific and non-word-specific. Non-word-specific features consisted of fixation duration, saccade lengths, short refixations, and the total number of fixations. On the other hand, the word-specific features contained gaze duration on each word and the number of revisits on each word. Based on the same dataset as Smyrnakis et al., Asvestopoulou et al. (2019) developed a tool called DysLexML. The classifier with the highest accuracy on noise-free data is linear SVM, used on features selected by LASSO regression at $\lambda 1SE$, which gave an accuracy of 87.87%, and up to 97%+ when using leave-one-out cross-validation. In recent years, Raatikainen et al. (2021) used a hybrid method consisting of an SVM classifier with random forest feature selection for dyslexia detection with data recorded from eye movement. The best-performing SVM model of their study scored an accuracy of 89.7%.

| Subj | Score | $n$ Texts | WPM | Age | Gender | Diagnosed |
|---|---|---|---|---|---|---|
| | | | Readers with dyslexia | | | |
| P23 | 1.00 | 2 | 200.0 | 33 | F | 16 |
| P24 | 0.80 | 2 | 203.7 | 64 | F | 9 |
| P25 | 0.82 | 4 | 142.0 | 20 | F | 16 |
| P26 | 0.57 | 2 | 86.7 | 32 | M | 12 |
| P27 | 0.71 | 4 | 137.4 | 53 | M | 48 |
| P28 | 0.93 | 4 | 173.3 | 25 | F | 15 |
| P29 | 0.73 | 3 | 143.3 | 25 | F | 21 |
| P30 | 0.93 | 4 | 179.0 | 61 | M | 50 |
| P31 | 0.75 | 2 | 61.9 | 20 | M | 15 |
| P33 | 0.86 | 2 | 59.3 | 30 | F | 8 |
| P34 | 0.62 | 2 | 107.4 | 56 | F | 9 |
| P35 | 0.71 | 4 | 285.1 | 24 | F | 19 |
| P36 | 0.40 | 2 | 58.5 | 23 | F | 11 |
| P37 | 0.58 | 4 | 270.7 | 25 | F | 23 |
| P38 | 0.75 | 2 | 115.5 | 30 | M | 29 |
| P39 | 1.00 | 1 | 160.2 | 32 | F | 17 |
| P40 | 0.92 | 4 | 173.3 | 29 | M | 7 |
| P41 | 0.88 | 4 | 154.9 | 51 | F | 50 |
| **AVG** | **0.78** (0.16) | **2.9** (1.1) | **150.7** (65.0) | **35.1** (14.7) | **67.7%F** | **20.8** (14.3) |
| | | | Readers without dyslexia | | | |
| **AVG** | **0.81** (0.11) | **4.4** (1.5) | **276.8** (54.6) | **30.7** (10.8) | **78% F** | – |

Table 1: Overview of readers with dyslexia included in the study. Average and standard deviations are in brackets. Score is the accuracy of the answers to the comprehension questions; Diagnosed refers to the age at which the participants were diagnosed with dyslexia. Aggregated data from the 18 readers without dyslexia from Hollenstein et al. (2022) for comparison.

## 3 Data Collection

Data acquisition follows Hollenstein et al. (2022), but the most important points are repeated here. The only procedural difference is the additional two reading tests administered to participants with dyslexia as described in §3.3.

### 3.1 Participant Selection

The participant selection for this study of natural reading is purposefully broad and follows the requirements for Hollenstein et al. (2022) from which we sample the typical readers. Prior to this, we excluded four participants from the non-dyslexic group from the analysis due to poor calibration or reported attention deficit disorder. The only difference to our participant sampling is that all dyslexic readers are officially diagnosed with dyslexia. There is no age limit and no required educational background but all participants are adults, and native speakers of Danish. All have normal vision or corrected-to-normal (glasses or contact lenses), but no readers included in the analysis had a known attention deficit disorder. All participants signed an informed consent and all digital data is pseudonymised. Due to the ab-sence of an official dyslexia diagnosis, we discard the data from one subject for further analysis but include 18 readers in the dyslexic group. Participant statistics for all included dyslexic participants are presented in Table 1 with a summary of the 18 non-dyslexic participants for comparison.

### 3.2 Reading Materials

We used the same set of reading materials as Hollenstein et al. (2022) presented in the same way. They are 46 transcribed and proofread Danish speeches, accessed from the Danske Taler archive (https://dansketaler.dk). Table 2 shows an overview. The readability of each speech was calculated from a LIX score, which is based on the length of the words and sentences in a text (Björnsson, 1968). Each reader read a subset of the full dataset reported in $n$ Texts in Table 1.

**Reading Comprehension Questions** To prevent mindless reading, comprehension questions were added to occur after approximately 20% of the paragraphs that contain more than 100 characters following Hollenstein et al. (2022). The average accuracy of the comprehension questions per participant can be seen in Table 1 in the Score

| | MIN | MAX | MEAN | STD | TOTAL |
|---|---|---|---|---|---|
| SENTS PER DOC | 37 | 134 | 92.4 | 29.4 | 1,849 |
| TOKENS PER DOC | 978 | 2,846 | 1,744.8 | 533.1 | 34,897 |
| WORD TYPES PER DOC | 391 | 1,056 | 603.6 | 159.4 | 7,361 |
| LIX PER DOC | 26.4 | 50.1 | 37.2 | 7.2 | – |
| FREQUENCY PER DOC | 0.68 | 0.79 | 0.74 | 0.03 | – |
| SENT LEN IN TOKENS | 1 | 119 | 10.8 | 15.9 | – |
| TOKEN LEN IN CHARS | 1 | 33 | 4.5 | 3.0 | – |

Table 2: Statistics on the 46 documents that comprise the reading material. TOTAL is the dataset total. LIX is the readability score. For typical readers, a text with a LIX score between 25 and 34 is considered easy, whereas a text scoring more than 55 is considered difficult and corresponds to an academic text. The frequency is measured by the proportion of words included in the 10,000 most common Danish words from https://korpus.dsl.dk/resources/details/freq-lemmas.html

column.

### 3.3 Lexical Assessment

All participants with dyslexia performed two lexical assessment tests, which are used as a control test for the current study. Both tests are developed by the Centre of Reading Research, University of Copenhagen. The purpose of the tests is to have a comparable benchmark for a lexical assessment unrelated to the eye movements of the participants with dyslexia.

Nergård-Nilssen and Eklund (2018) found in their psychometric evaluation that a pseudohomophone test is of high reliability and that such a test incorporates evaluations that provide accurate discrimination of readers with dyslexia. Due to this finding, as well as the fact that the pseudohomophone task is used in the Danish dyslexia test, a pseudohomophone test was selected as one of the lexical assessment tests for the current study. For the sake of reliability and providing insightful findings on reading skills, a reading comprehension test was also used as a complementary lexical assessment test.

**Reading Comprehension Test** The original purpose of the reading comprehension test[5] is to provide easy access for adults to receive an informal evaluation of their reading skills, and to stress that more adults are seeking help with developing their reading skills (Jensen et al., 2014). It takes ten minutes to complete, making it relatively short, yet insightful. The tasks in the test consist of three variants of cloze tests, which are tests where the participants must select a missing word in a sentence, e.g., It had been raining for some ______ [days, moments, countries] (our translation).

As the reading task is an online self-assessment test that requires no log-in or external assistance, requirements, or access, the participants without dyslexia in the experiment were contacted after their participation in the eye-tracking experiment to voluntarily take the test at home to serve as a control group. Ten participants without dyslexia submitted their scores as a contribution to this experiment.

The aggregated results for both reader groups are presented in Table 3. We observe that readers with dyslexia generally have a lower score and a larger variance. A two-tailed t-test showed that this difference is significant ($p < 0.001$).

| GROUP | $n$ | MEAN | MIN | MAX |
|---|---|---|---|---|
| DYSLEXIC | 18 | 3.5 | 0.7 | 5.2 |
| NOT DYSLEXIC | 10 | 5.7 | 4.4 | 7.1 |

Table 3: Reading task scores for participants of both reading groups. A score between 0–3.4 indicates that the reader may find many texts difficult and time-consuming to read, and a score between 3.5–3.9 indicates that the reader may find some texts difficult and/or time-consuming to read. A score over 4 indicates good reading skills.

**Pseudohomophone Test** The second linguistic assessment we conducted with the participants with dyslexia was a pseudohomophone [6] and was developed as a part of a diagnostic reading test for adults. The test encompasses 38 tasks where each task consists of four non-words, of which one of the words sounds like a real Danish word when pronounced. The difficulty of the 38 tasks increases gradually. The participants get five minutes to complete as many tasks as possible. Knowledge of the words of the test is required to perform it, but as the words are frequent, everyday words in Danish, it is assumed that native, adult readers are familiar with the words. Translated examples of the words are: cheese, eat, steps, factory, and help.

---

[5]Accessed from https://selvtest.nu/

[6]Accessed from https://laes.hum.ku.dk/test/

| Group | $n$ | Acc |
|---|---|---|
| NO READING DIFFICULTIES | 72 | 66% |
| IN PROGRAMS FOR DYSLEXIC STUDENTS | 46 | 23% |
| IN LITERACY READING PROGRAMS | 167 | 31% |
| COPCO READERS WITH DYSLEXIA | 18 | 33% |

Table 4: Pseudohomophone test accuracies. The three top rows are standards from the official documentation of the test material for comparison.

The result is presented in Table 4 compared to standard scores from the documentation of the test[7]. We observe that the scores from the readers with dyslexia in the current study are on par with the standard scores of adults in literacy reading programs and higher than the standards for adults in programs for dyslexic readers. However, all quartile scores for our group of readers with dyslexia are about half compared to the standards for adults without reading difficulties.

## 3.4 Experiment Procedure

Eye movement data were collected with an infrared video-based EyeLink 1000 Plus eye tracker (SR Research) and follow Hollenstein et al. (2022). The experiment was designed with the SR Experiment Builder software. Data is recorded with a sampling rate of 1000Hz. Participants were seated at a distance of approximately 85 cm from a 27-inch monitor (display dimensions 590 x 335 mm, resolution 1920 x 1080 pixels). We recorded monocular eye-tracking data of the right eye. In a few cases of calibration difficulties, the left eye was tracked.

A 9-point calibration was performed at the beginning of the experiment. The calibration was validated after each block. Re-calibration was conducted if the quality was not good (worst point error $< 1.5°$, average error $< 1.0°$).Drift correction was performed after each trial, i.e. each screen of text. Minimum calibration quality measure of the recording ("good" calibration score, or "fair" in exceptionally difficult cases).

**Experiment Protocol** Participants read speeches in blocks of two speeches. The experiment was self-paced meaning there were no time restrictions. Thus, the participants read in their own pace for comprehension which is what we dub 'natural reading'. Between blocks, the

participants could take a break. Each participant completed as many blocks as they were comfortable within one session. The order of the blocks and the order of the speeches within a block were randomized. Instructions were presented orally and on the computer screen before the experiment started. All participants first completed a practice round of reading a short speech with one comprehension question. The experiment duration was between 60 and 90 minutes.

**Stimulus Presentation** The text passages presented on each screen resembled the author's original division of the story into paragraphs as much as possible. Comprehension questions were presented on separate screens. The text was in a black, monospaced font (type: Consolas; size: 16pt) on a light-gray background (RGB: 248,248,248). The texts spanned max. 10 lines with triple line spacing. We used a 140 pixels margin at the top and bottom, and 200 pixels side margin for a screen resolution of 1920x1080.

## 4 Data Processing

### 4.1 Event Detection

This procedure also follows Hollenstein et al. (2022) closely. During data acquisition, the eye movement events are generated in real-time by the EyeLink eye tracker software during recording with a velocity- and acceleration-based saccade detection method. The algorithm defines a fixation event as any period that is not a saccade or a blink. Hence, the raw data consist of (x,y) gaze location coordinates for individual fixations.

We use the DataViewer software by SR Research to extract fixation events for all areas of interest. Areas of interest are automatically defined as rectangular boxes surrounding each text character on the screen, as shown in Figure 1. For later analysis, only fixations within the boundaries of each displayed character are extracted. Therefore, data points distinctly not associated with reading are excluded. We also set a minimum duration threshold of 100ms.

### 4.2 Feature Extraction

In the second step, we use custom Python code to map and aggregate character-level features to word-level features. These features cover the reading process from early lexical access to later syntactic integration. The selection of features is

---

[7]https://laes.hum.ku.dk/test/find_det_der_lyder_som_et_ord/standarder/

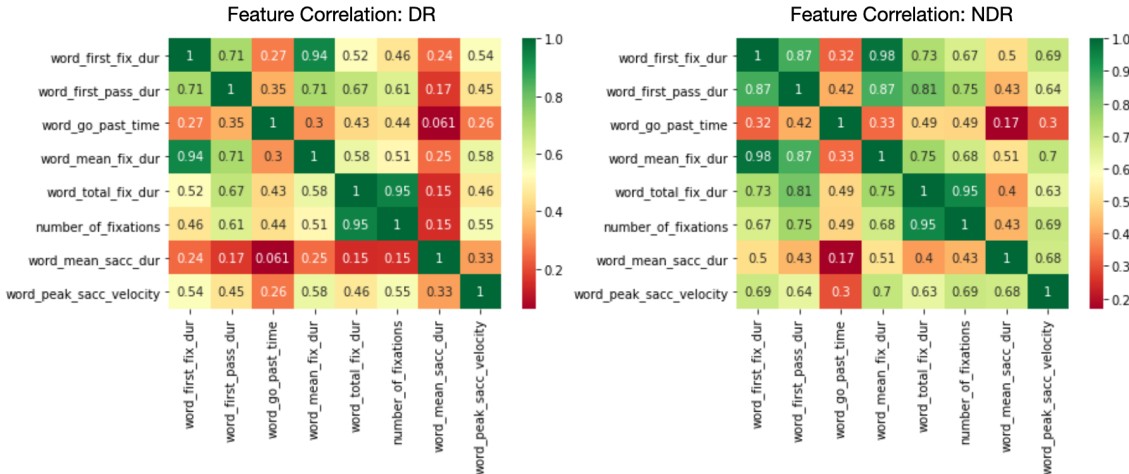

Figure 2: Correlation matrices showing correlations between all features recorded from readers with dyslexia (DR; left) and readers without dyslexia (NDR; right).

inspired by similar corpora in other languages (Siegelman et al., 2022; Hollenstein et al., 2018; Cop et al., 2017) as well as features known to show strong effects in eye movements from readers with dyslexia (Biscaldi et al., 1998; Pirozzolo and Rayner, 1979; Rayner, 1986). We extract the following eye-tracking features:

- nFIX: The total number of fixations on the current word.

- FFD: Duration of the first fixation of the current word.

- MFD: Mean duration of all fixations on the current word.

- TFD: Total fixation duration on the current word.

- FPD: first pass duration, The summed duration of all fixations on the current word prior to progressing out of the current word (left or right).

- GPT: go-past time, the sum duration of all fixations prior to progressing to the right of the current word, including regressions to previous words that originated from the current word.

- MSD: mean saccade duration, Mean duration of all saccades originating from the current word.

- PSV: peak saccade velocity, Maximum gaze velocity (in visual degrees per second) of all saccades originating from the current word.

The feature correlations for readers with and without dyslexia are shown in Figure 2. They illustrate that the correlation of the features is generally higher for readers without dyslexia compared to those with dyslexia. This may indicate that the data varies more among readers with dyslexia, suggesting that the reading pattern of the participants with dyslexia includes greater variability. The highest correlated features are those related to fixations, with the highest correlated pairs being first fixation duration and mean fixation duration, as well as total fixation duration and the number of fixation duration. A t-test analysis was performed to compare the features recorded from readers with and without dyslexia, revealing that all eight features show a significant difference between groups ($p < 0.0001$).

## 5 Dyslexia Classification

We experiment with three types of classifiers using features on two different levels of aggregation; sentence-level and trial-level. A trial corresponds to the text presented on a single screen, roughly corresponding to paragraphs from the original text materials. For both levels of aggregation, the eye-tracking features of each word in a sentence or trial, respectively, are averaged to get a single vector of eight features for each sample. Further, we experiment with adding standard deviations (+STD) and max values (+MAX). Therefore, we

| | $n$ SAMPLES | |
|---|---|---|
| EXPERIMENT TYPE | NON-DYSLEXIC | DYSLEXIC |
| TRIAL-LEVEL | 5,147 | 4,144 |
| SENTENCE-LEVEL | 21,859 | 17,477 |

Table 5: Dataset size.

train classifiers, where each sample corresponds either to the eye-tracking information from a sentence or from a full trial. Dataset sizes are presented in Table 5. The data is split into 90% training data and 10% test data. We use an additional 10% of the training data as a validation split for the Long Short-Term Memory (LSTM). For all experiments, we randomly undersampled the non-dyslexic datasets for training, but not testing. We perform 5 runs taking different random samples from the data of readers without dyslexia and report the average performance.

**SVM and Random Forest Classifiers** The eye-tracking features are normalised with a min-max scaler that gives each instance a number between 0 and 1.We use a grid search to tune the hyper-parameters of both SVM (the best regularization parameter $C = 100$) and random forest (the best maximum depth=9, and the optimal number of estimators=200) in a 5-fold cross-validation setup on the full train set. The classifiers are implemented with the scikit-learn library for Python. The SVM uses a linear kernel. In addition to taking the mean feature values per word or trial (i.e., aggregating the eye-tracking features of all individual words), we also experiment with adding the standard deviations and maximum values of each feature.

**LSTM Classifiers with Sequential Word Features** We train a recurrent neural network optimized for sequential data, namely an LSTM. As LSTMs perform well with sequences and data consisting of large vocabularies and are effective in memorizing important information, it can be beneficial to dyslexia detection to predict the probability of class for a sentence, given the observed words. Therefore, the inputs for the LSTM network are the same eye-tracking features, but rather than aggregating on the full trial or sentence, each word is assigned a feature vector. The sequences were then padded to the maximum sentence or trial length, respectively. We use two LSTM layers, with 32 and 16 dimensions, respectively, and a dropout rate of 0.3 after the first layer. Fi-

nally, we use a sigmoid activation function for outputting the probabilities of each class. The models are trained with a batch size of 128, using a cross-entropy loss and a RMSprop optimizer with a learning rate of 0.001. We implement early stopping with a patience of 70 epochs on the maximum validation accuracy and save the best model. The model was implemented using Keras.

| MODEL | TRIAL | SENTENCE |
|---|---|---|
| SVM | 0.80 (0.018) | 0.71 (0.004) |
| SVM + STD | 0.81 (0.010) | 0.71 (0.006) |
| SVM + STD + MAX | 0.81 (0.014) | 0.72 (0.007) |
| RF | 0.83 (0.012) | 0.72 (0.001) |
| RF + STD | **0.85** (0.015) | 0.72 (0.007) |
| RF + STD + MAX | **0.85** (0.010) | **0.73** (0.006) |
| LSTM | 0.82 (0.030) | 0.71 (0.037) |

Table 6: Average F1 score (standard deviation across five runs in brackets) for SVM, R(random)F(orest) and LSTM.

## 5.1 Results

The trial-level and sentence-level results for the dyslexia classification task are presented in Table 6. We observe that trial-level classifiers achieve much higher results than sentence-level classifiers, which is to be expected since the latter includes reading data from fewer words. However, for the SVM and random forest, the features are aggregated. Hence there will be an upper limit of text length suitable for these methods. The random forest achieves the best results on both levels and a wider range of features (namely, including standard variation and maximum value features) yields higher scores. The LSTM model does not outperform the simpler and faster-to-train random forest models and shows a higher variance between runs.

### 5.1.1 Misclassifications

To further analyze these results, we look at the confusion matrix and misclassified participants from the best model, namely the random forest classifier including mean, standard deviation, and maximum value features. The confusion matrices in Figure 3 show that more mistakes are made classifying samples from readers with dyslexia than from readers without dyslexia. This is more apparent at sentence-level where the number of samples is substantially larger.

Furthermore, we hypothesize that the classifier struggles to correctly classify samples from read-

ers with dyslexia that have reading patterns comparable to readers without dyslexia. The samples that are misclassified most frequently belong mostly to the same group of participants, both at sentence-level and at trial-level. The most frequently misclassified samples from readers with dyslexia were P28, P35, P23, P40, and P37 (in descending order of the number of misclassifications). We correlate the number of misclassified samples for all participants with dyslexia with their demographic and lexical text information and find a significant correlation between misclassifications and words per minute ($\rho = 0.79, p < 0.001$) and between misclassifications and reading comprehension scores ($\rho = 0.71, p < 0.001$). However, the correlation between misclassifications and pseudohomophone test scores is minimal and not significant. This shows that samples from readers with dyslexia with higher reading speed and better reading comprehension are more likely to be misclassified since the features are more similar to readers without dyslexia.

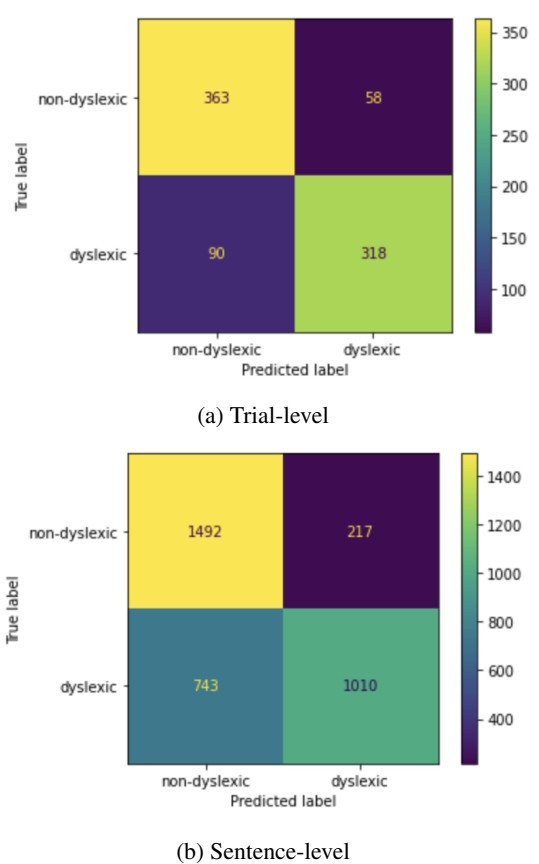

(a) Trial-level

(b) Sentence-level

Figure 3: Confusion matrices for the best classifier, RF+SDT+MAX, for each experiment level.

# 6 Discussion & Conclusion

We presented a dataset of eye-tracking recordings from natural reading from adults with dyslexia, which complements the CopCo dataset of readers without dyslexia (Hollenstein et al., 2022). Additionally, to the best of our knowledge, we presented the first attempt to predict dyslexia from eye-tracking features using Danish as a target language. The best-performing classifier of the current study achieves an F1 score of 0.85, using a random forest classifier trained with a feature combination that includes the aggregation of means, standard deviations, and maximum values of eight eye-tracking features.

While the recorded eye-tracking features proved to reflect vital information about the reading mechanisms of the participants, there were a considerably high number of misclassifications of fast and skilled readers with dyslexia. This indicates that a fast reading speed is atypical for a reader with dyslexia. These results contribute to findings that the symptoms of dyslexia occur in varying degrees and thus underline the importance of developing a reliable assessment tool for dyslexia that can reduce the number of misclassifications.

Moreover, due to known comorbidities across reading disorders (Mayes et al., 2000) that can be reflected in eye movements (e.g., attention and autism spectrum disorders), as the dataset continues to grow, we will include these populations of readers in the data collection to learn to classify different subgroups readers correctly.

Precise criteria for dyslexia diagnosis remain difficult to standardise with the varying degrees of the symptoms and indicators of the disorder, which is why the condition deserves more attention. As eye-tracking recordings provide insightful information about cognitive processes in naturalistic tasks such as reading, they can be a beneficial tool for dyslexia prediction. Eye tracking can be a stepping stone to achieving more reliable screening methods for dyslexia.

## Acknowledgments

Maria Barrett is supported by a research grant (34437) from VILLUM FONDEN.

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
