# OpenReview forum: "Dyslexia Prediction from Natural Reading of Danish Texts"
_NoDaLiDa/2023/Conference — NoDaLiDa 2023_

### Official Review · Reviewer_z7Y1 · 2023-03-10
**Some interesting eye-tracking-based dyslexia prediction systems**

**Rating:** 7
**Confidence:** 4

**Review:**

The paper presents various systems built to detect dyslexia from eye-tracking data.

Strengths:

* As the paper clearly states, most effort is spent on detecting dyslexia at a young age. In contrast, the study focuses on adults, thus filling an important gap in the research area.
* The best system achieves quite good results (0.85 F1), and could potentially become an effective tool in screening.
* The models were evaluated both at the sentence and trial level.

Weaknesses:
* Although the authors report sentence and trial-level results, it would be interesting to see subject-level evaluations. This would give readers a sense of how well the proposed solution would work, and how many sentences/trials are needed per person to get an accurate prediction.
* Regarding the data collection procedure, I understand that the current study focuses purely on dyslexia, but excluding people with attention deficits is a major limitation of the current solution. It means that the system is not universal, and in the future, the authors should focus on extending their solution to handle various other traits of the subjects that might affect their reading skills (as mentioned in conclusion).
* In section 3.4, the authors mention that "Re-calibration was conducted if the quality was not good (worst point error < 1.5◦, average error < 1.0◦)". This means that an average error of almost 1.0◦ was still acceptable, which results in a large circular area containing the true focus-point (by my calculations, its radius is around 49 pixels, and considering that the font-size was 16 it could cover multiple short words). Figure 1 seemingly contradicts this (I might be wrong in my calculations), or do the circles in that figure represent the time focused on the given area? Overall, I think the inaccuracy of the recording device should be disclosed in a discussion section.
* Regarding the feature extraction, I am curious how it was handled when a reader's focus was in between words (as the features are on a word-level)? Additionally, the features are purely composed of the eye-tracing recordings, but I think it would be beneficial to include some information about the word itself  (e.g. length, part-of-speech tag, or even a word embedding vector) as not all words can be read with the same ease, and some keywords might require more focus to understand the sentence properly.
* Lastly, I noticed that the data used in the study has a considerable bias towards female readers (~70% of the participants are F in Table 1). Have the authors assessed if their solutions have some gender bias (i.e. better performance for female readers than for males)?

Overall I think the paper is interesting from an academic standpoint, but there is plenty of work ahead before the models can be integrated into the dyslexia screening procedure.

**Paper Type:**

Long paper

---

### Official Review · Reviewer_UCwr · 2023-03-10
**Eye-tracking dyslexia in Danish**

**Rating:** 9
**Confidence:** 5

**Review:**

The paper presents work where eye-tracking is used to detect dyslexia in Danish readers.  It describes how an eye-tracking database of dyslexic and non-dyslexic readers is gathered and analysed and how the data is used to develop a dyslexia classifier based on SVM, Random Forests and LSTM networks.  The work is well presented in a clearly written paper which should be accepted to the conference.

Two minor comments:
  It is an interesting hypothesis that the non-transparent orthography in Danish makes it easier to detect dyslexia.  This hypothesis is not addressed in the paper and I don't see why it necessarily holds since non-transparent orthography would affect both dyslexic and non-dyslexic readers alike.  This is something that could be investigated (with experiments spanning many languages) but since it is not the topic of the paper, I would not make such sweeping statements.

In Section 5 you say "For all experiments, we randomly under-sampled the non-dyslexic datasets for training, but not testing. "  I stumbled over this for a while thinking, how did you estimate the false positive rate if you didn't include non-dyslexic data in your testing, only to realise that it is only the under-sampling (which makes sense) that you are referring to.  Maybe you can rephrase this to avoid this kind of misunderstanding.

**Paper Type:**

Long paper

---

### Official Review · Reviewer_ik6e · 2023-03-11
**Solid paper on dislexia prediction from eye-tracking data**

**Rating:** 7
**Confidence:** 4

**Review:**

The paper presents machine learning experiments on eye-tracking data designed to detect dyslexia from natural reading. The paper relies on a recently released dataset for Danish (Hollenstein 2022). I found the paper to be very well-written, interesting, and the experimental design is sound.

A few comments:
1) what is "natural reading"? Does it simply refer to reading a text without having predefined pieces of information / questions to look for?
2) On page 2, what exactly is the Hellerup Test? Is this a subpart of the Dyslexia Test described above? Or something else?
3) If I understood correctly the experimental design, the train/test split is done at the level of trial or sentences. But this means that data from the test and train sets might come from the same human reader. This might distort the results, for instance if a human reader is associated with some peculiar eye-tracking data which is "picked up" by the model during training. I would suggest to split the train, development and test sets on the basis of participants instead of trials of sentences. Furthermore, to obtain more reliable results, I would also suggest to perform k-fold cross validation (again, on the basis of a split on the participants) instead of relying on a fixed split.
4) In addition to SVMs and Random Forests, I would also suggest to test a logistic regression and/or a decision tree, as it would provide explainable models that would be useful to investigate which features are most useful.
5) Regarding the sequential model, I would also suggest to test time-series models in addition to the LSTM (see for instance the "sktime" library for a list of possible models).
6) The experiments are aimed at predicting the presence of dyslexia at the sentence or trial level. However, in real-world settings, the goal of such a test is to predict dyslexia for a given person based on the series of trials conducted by that person. It would be useful to discuss how such prediction could be done in practice. For instance, one could take the mean or max probability of dyslexia across all trials/sentences. Or one could train a statistical model that would directly perform the prediction based on the collection of eye-tracking data for all trials. In any case, it would be useful to discuss how the prediction could be done at the person-level, since this is the ultimate goal of those tests.
7) Last but not least, I have a more general interrogation. The experiments compare the results of the machine learning to a "ground truth", in this case whether the readers are officially diagnosed with dyslexia or not. But this ground truth is most likely a biased indicator (this diagnosis is, after all, also derived from prior tests and analyses that are themselves imperfect). How sure are we that the false positives of the ML models do not have undiagnosed dyslexia (although perhaps in a milder form)? And how sure are we that the false negatives actually are false negatives, and not due to an erroneous diagnosis for those individuals? If there is a non-negligible probability of errors/omissions in this "ground truth", perhaps one could analyze the results in terms of inter-annotator agreement (between the official diagnosis and the outcome of the ML model using eye-tracking data) instead of F1 score. In any case, a discussion of those questions would be welcome.


**Paper Type:**

Long paper

---

### Decision · Program_Chairs · 2023-03-17

Accept